# Immune Response after SARS-CoV-2 Infection with Residual Post-COVID Symptoms

**DOI:** 10.3390/vaccines11091413

**Published:** 2023-08-24

**Authors:** Tanyaporn Pongkunakorn, Thamonwan Manosan, Apinya Surawit, Suphawan Ophakas, Pichanun Mongkolsucharitkul, Sureeporn Pumeiam, Sophida Suta, Bonggochpass Pinsawas, Nitat Sookrung, Nawannaporn Saelim, Kodchakorn Mahasongkram, Pannathee Prangtaworn, Anchalee Tungtrongchitr, Watip Tangjittipokin, Suthee Mangmee, Kobporn Boonnak, Tassanee Narkdontri, Nipaporn Teerawattanapong, Rungsima Wanitphadeedecha, Korapat Mayurasakorn

**Affiliations:** 1Siriraj Population Health and Nutrition Research Group, Research Department, Faculty of Medicine Siriraj Hospital, Mahidol University, Bangkok 10700, Thailand; tanyaporn.pon@mahidol.edu (T.P.); thamonwan.manosan@gmail.com (T.M.); apinya.sua@mahidol.ac.th (A.S.); phakhaun1234@gmail.com (S.O.); pichanun.mon@mahidol.edu (P.M.); sureeporn.pum@mahidol.edu (S.P.); sophida.sut@mahidol.edu (S.S.); bonggochpass.pin@mahidol.edu (B.P.); 2Center of Research Excellence on Therapeutic Proteins and Antibody Engineering, Department of Parasitology, Faculty of Medicine Siriraj Hospital, Mahidol University, Bangkok 10700, Thailand; nitat.soo@mahidol.ac.th (N.S.); nawannaporn.sae@mahidol.edu (N.S.); kodchakorn.mah@mahidol.ac.th (K.M.); pannathee.pra@mahidol.ac.th (P.P.); anchalee.tun@mahidol.ac.th (A.T.); 3Department of Immunology, Faculty of Medicine Siriraj Hospital, Mahidol University, Bangkok 10700, Thailand; watip.tan@mahidol.edu (W.T.); suthee.mag@gmail.com (S.M.); kobporn.boo@mahidol.edu (K.B.); 4Research Department, Faculty of Medicine Siriraj Hospital, Mahidol University, Bangkok 10700, Thailand; tassanee.nar@mahidol.edu (T.N.); nipaporn.tee@mahidol.edu (N.T.); 5Department of Dermatology, Faculty of Medicine Siriraj Hospital, Mahidol University, Bangkok 10700, Thailand; rungsima.wan@mahidol.ac.th

**Keywords:** long COVID, neutralizing antibody, breakthrough infection, immunogenicity, Omicron

## Abstract

Many patients develop post-acute COVID syndrome (long COVID (LC)). We compared the immune response of LC and individuals with post-COVID full recovery (HC) during the Omicron pandemic. Two hundred ninety-two patients with confirmed COVID infections from January to May 2022 were enrolled. We observed anti-SARS-CoV-2 receptor-binding domain immunoglobulin G, surrogate virus neutralization test, T cell subsets, and neutralizing antibodies against Wuhan, BA.1, and BA.5 viruses (NeuT). NeuT was markedly reduced against BA.1 and BA.5 in HC and LC groups, while antibodies were more sustained with three doses and an updated booster shot than ≤2-dose vaccinations. The viral neutralization ability declined at >84-days after COVID-19 onset (PC) in both groups. PD1-expressed central and effector memory CD4^+^ T cells, and central memory CD8^+^ T cells were reduced in the first months PC in LC. Therefore, booster vaccines may be required sooner after the most recent infection to rescue T cell function for people with symptomatic LC.

## 1. Introduction

Severe acute respiratory syndrome coronavirus 2 (SARS-CoV-2) is constantly changing and accumulates mutations in its genetic code over time [1]. The Omicron lineage (B.1.1.529), recognized as a variant of concern in November 2021, spread in an unprecedented manner to become dominant worldwide and later evolved into numerous sub-lineages such as BQ.1, BA.2.75, BA.5 and BA.4 [2,3]. These sub-lineages share many genetic mutations with earlier variants. Omicron displaced Delta as the predominant variant during the study period. Randomly selected SARS-CoV-2 variants captured during surveillance by the Department of Medical Science, Thailand; the UK Health Security Agency (UKHSA); and the Centers for Disease Control and Prevention, USA during weeks 1 to 4 of 2023 demonstrated that almost all new infections in Thailand [4], the U.K. [5] and the U.S. [6] were due to Omicron BA.5/BA.4, BQ.1, BA.2.75, and XBB.1.5.

In many countries, transmission of the BA.5 and BA.4 variants led to biological changes. The mutations in the variants accelerate infectivity in both fully vaccinated and boosted individuals and in people who had been immunized to earlier forms of Omicron and other variants with unique mutations in the spike protein, resulting in a wide range of clinical presentations. Most individuals with breakthrough SARS-CoV-2 infection experience asymptomatic, mild-to-moderate, or severe coronavirus disease 2019 (COVID-19) [7,8]. A previous study [1,2] showed that sera from previous pre-Omicron, Omicron BA.1 or BA.2 and BA.5/BA.4 breakthrough infection exhibited significant reduction in neutralization against BA.5/BA.4 and BA.2.75, thereby raising the possibility of Omicron reinfection. These results may reflect a combination of immune evasion and a gradual decline in immunity.

Several lines of evidence suggest that there are multi-systemic humoral and immune responses during acute illness and after recovery from the acute phase of COVID-19 infection. These factors contribute to the host defense and pathogenesis of severe COVID-19 and residual post-COVID syndrome (long COVID (LC)). In the acute phase, marked immune dysregulation with lymphocytopenia, increased neutrophils, and activation of the coagulation cascade has been described [7,9]. After acute COVID-19 infection, 10–70% of patients reported physical and mental symptoms 2–3 months after infection [10,11,12]. Many hospitalized symptomatic COVID-19 patients have experienced symptoms of LC. In addition, more than half of patients with prior mild-to-moderate COVID-19 have reported LC that persisted more than 4 months after infection [13,14]. LC manifests as a diversity of symptoms affecting various organs, as seen in COVID-19 patients who did not recover fully [15]. The most predominant LC symptoms are headache, breathlessness, cough, chest pain, abdominal symptoms, myalgia, fatigue, cognitive difficulties, anxiety, depression [16,17], and neurological and psychiatric symptoms [18]. T cells and persistent immune dysfunction have been shown to be associated with LC even after mild COVID [19]. For example, expression of effector molecules in memory T cells was reduced in neurological patients with LC symptoms, and a sustained increase in T cell activity was observed in this group in response to SARS-CoV-2 mRNA vaccination compared with healthy COVID convalescents (HC) [20]. Further investigation and verification is needed to establish the drivers of immune dysfunction and immune response after vaccination and to identify the specific effects on LC.

## 2. Materials and Methods

This retrospective cohort study compared immunogenicity after COVID-19 between individuals with residual LC symptoms and those with HC in terms of pathophysiology, immunology, and clinical outcome. In addition, immune responses were determined at 4 weeks after a subset of these participants were voluntarily administered an additional 30 μg of BNT162b2 (Pfizer).

### 2.1. Study Population

All patients (age: 18–65 years) were invited to voluntarily participate in our study of immune response to SARS-CoV-2 after breakthrough infection. Eligible participants were individuals (1) whose medical records confirmed that they had SARS-CoV-2 infection between January and May 2022 (during the Omicron-dominant pandemic) and (2) who had not received additional COVID-19 vaccinations after infection. Pregnant participants and people with a history of allergic reactions after either COVID-19 vaccination or medication were excluded. Participants were defined as LC if they had residual self-reported LC symptoms lasting longer than 1 month PC. LC symptoms were based on the presence of at least 1 of the 3 primary symptoms: fatigue/myalgia, shortness of breath, and anorexia [19]. Patients without LC symptoms were designated as “healthy COVID-19 recovery; HC”. Participants in the 2 groups were matched for sex and age (±5 years). On the basis of their residual symptoms reported in the questionnaires, 158 participants with LC symptoms and 134 HC participants were included. We performed a subset of analyses on an independent group of 36 post-acute COVID-19 patients (outpatient adults infected during the Omicron BA.5/BA.4 wave) to compare their immune responses with our primary cohort. Prior infections were classified as Omicron BA.1/BA.2 dominant versus Omicron BA.5/BA.4 dominant, depending on whether the infections occurred before or after the BA.5/BA.4 wave (which began in Thailand in June 2022) [4].

### 2.2. Booster Doses and Immunogenicity

In the study cohort, a subset of 100 patients (age ≥ 18 years) were voluntarily recruited for an immunogenicity follow-up study after receiving 30 μg of BNT162b2 (Pfizer) as a booster shot at 29–84 days PC. All participants provided informed consent. The participants were tested for SARS-CoV-2 antibodies and surrogate virus neutralization (sVNT) against the SARS-CoV-2 Wuhan and Omicron variants. Those who declined the booster vaccination were asked to retest IgG before the next booster vaccination to determine the waning of their immunity. The Department of Disease Control in the Ministry of Public Health (Thailand) kindly provided the Pfizer-BioNTech (Comirnaty) BNT162b2 vaccine (Lot. 1L085A). This RNA vaccine is part of a state-of-the-art approach that uses genetically modified mRNA of viral spike proteins embedded in lipid nanoparticles. After intramuscular injection into the human body, the mRNA enables ribosomes in cells to synthesize viral spike proteins that safely elicit an immune response [21]. We studied the immunogenicity of this boost 4 weeks after administration of the booster shot.

### 2.3. Omicron BA.5/BA.4 Immunogenicity

The pilot study recruited 36 patients diagnosed with COVID-19 between 15 July and 14 September 2022. This subgroup of patients provided a proxy for BA.5/BA.4 infections that could be used to determine the frequency of these variants during the recruitment period. Immune responses against all Omicron variants were assessed for viral neutralization against BA.5 and BA.4 [4].

### 2.4. Outcome Measures

The primary outcome was a comparison of baseline clinical and biological characteristics of patients in the LC and HC groups. Outcomes included demographic, clinical, and biochemical data. Participants provided study personnel with a copy of their state immunization certificates to verify their immunization status. The “date of illness onset” was the day on which self-reported new respiratory symptoms appeared. Viral load was included in cycle threshold analyses by vaccine exposure group and self-reported symptoms, as previously described [7].

### 2.5. SARS-CoV-2 Real-Time Quantitative Reverse Transcription Polymerase Chain Reaction

The COVID-19 diagnosis was based on the detection of ≥2 SARS-CoV-2 genes in nasopharyngeal swabs, throat swabs, or other respiratory specimens by real-time quantitative reverse transcription polymerase chain reaction (RT-qPCR), as previously described [22]. The COVID-19 diagnostic assay used a qualitative RT-PCR probe. An Allplex 2019-nCoV Assay (Seegene, Seoul, Republic of Korea) was used to detect SARS-CoV-2. The COVID-19 genes detected were the nucleocapsid (N), envelope (E) of Sarbecovirus, and RNA-dependent RNA polymerase (RdRp) of COVID-19, according to the manufacturer’s instructions and described previously [23]. Positive results from an antigen test kit were also accepted as confirmation of COVID-19 in cases where RT-PCR could not be evaluated.

### 2.6. Serological Assays

Blood samples were collected at baseline (pre-booster) and 4 weeks after booster shot (after boost) from a subset of participants from both groups. Briefly, plasma samples were assayed for anti-SARS-CoV-2 receptor-binding domain immunoglobulin G (anti-RBD IgG; SARS-CoV-2 IgG II Quant for use with Architect; Abbott Laboratories, Abbott Park, IL, USA), as described previously [24]. This assay linearly measures antibody levels between 21.0 and 80,000.0 arbitrary units (AU)/mL. The value was then converted to the WHO international standard concentration as binding antibody units per mL (BAU/mL) with the equation provided by the manufacturer (BAU/mL = 0.142 × AU/mL). Values greater than or equal to the cutoff of 50 AU/mL or 7.1 BAU/mL were considered seropositive.

A surrogate virus neutralization test (sVNT) was performed against wild-type (Wuhan) and the Omicron (B.1.1.529) strains. Briefly, plasma was pre-incubated with horseradish peroxidase-conjugated receptor-binding domain protein (HRP-conjugated RBD protein). The mixture was transferred to wells containing streptavidin bound with biotin-conjugated angiotensin-converting enzyme 2 (ACE2). Finally, optical density absorbance was measured using a spectrophotometer at 450 nm. The sample diluent was used as a negative control. The inhibition rate was calculated according to the following formula:$$Inhibition\;rate\;\left(\%\right)=\left(1-\frac{{OD}_{450}\;of\;Sample}{\;{OD}_{450}\;of\;Negative\;control\;}\right)\times 100.$$

A microneutralization assays using authentic SARS-CoV-2 viruses (NeuT), SARS-CoV-2 Wuhan/WA1/2020, Omicron BA1, Omicron BA2, and Vero-E6 cells were kindly provided by Professor Florian Krammer and Dr. Juan Manuel Carreno Quiroz, Icahn Scholl of Medicine at Mount Sinai, New York, USA. The BA.5 variant was isolated from clinical samples collected at Siriraj Hospital Mahidol University using the Vero E6 cell line. The spike gene of BA.5 was confirmed by sequencing. Viruses were passaged in Vero-E6 cells and stored at −80 °C. Serum was heat inactivated at 56 °C for 30 min. The heat inactivated sera were serially diluted 1:10 to 1:10,240 before adding 100 TCID_50_ of SARS-CoV-2 in MEM media containing 0.5% BSA and incubated at room temperature for 1 h. Residual virus infectivity in the serum/virus mixture was assessed in quadruplicate wells of Vero E6 cells incubated in serum-free media at 37 °C and 5% CO_2_, and the viral cytopathic effect was read on day 5. The neutralizing antibody titer was calculated using the Reed–Muench method [25,26,27].

### 2.7. Peripheral Blood Mononuclear Cells Isolation and Collection

Peripheral blood was drawn into a tube supplemented with sodium heparin as anticoagulant [28]. Briefly, the blood sample was diluted 1:1 with phosphate-buffered saline, pH 7.2 (PBS; HyClone, Logan, UT, USA) and PBMCs were isolated by using Ficoll gradient centrifugation. Three mL of a density gradient medium (Lymphoprep; Stemcell Technologies, Cologne, Germany) were added to the bottom of a 15 mL tube. Then, 10 mL of the diluted blood samples were carefully layered on top and centrifuged at 400× *g* for 30 min at 25 °C (Allegra X-15R centrifuge; Beckman Coulter, Fullerton, CA, USA) using a swinging bucket rotor to separate the upper and lower fractions. Most of the upper layer was aspirated, leaving the mononuclear cells at the interphase. The peripheral blood mononuclear cells were transferred to another 15 mL tube, and 10 mL of PBS was added, mixed, and centrifuged at 40× *g* for 5 min at 25 °C. The supernatant was removed entirely, and the cells were resuspended in a small volume of PBS. When most of the platelets were removed, the cells were suspended in complete cryoprotective media with 10% dimethyl sulfoxide (DMSO; Sigma Life Science, Burlington, MA, USA) and fetal bovine serum (Gibco FBS; Thermo Fisher Scientific Inc., Waltham, MA, USA) in cryovials. Finally, the cryovials were frozen at −80 °C for 1 day before transfer to liquid nitrogen and storage.

### 2.8. T Cell Immunophenotyping Analysis

The cryopreserved PBMC vials were placed in a 37 °C water bath for rapid thawing. Then, thawed PBMCs (1 mL) were slowly added to 10 mL of warmed complete RPMI. The cells were then centrifuged at 400× *g* for 5 min. The supernatant was poured off. Two mL of complete RPMI was added to the pellet and gently mixed to suspend the cell pellet. The cell suspension was transferred to a 6-well tissue culture plate and incubated at 37 °C in a CO_2_ water jacketed incubator (37 °C, 5% CO_2_, 95% humidity), overnight. The PBMCs were harvested the next day. The number and viability of PBMCs were assessed by Trypan blue exclusion assay. PBMCs were then washed twice with dPBS. PBMCs were adjusted to 1–10 × 10^6^ cells in 100 μL of diluted Zombie aqua dye (BioLegend, San Diego, CA, USA) and incubated at room temperature for 15 min. The stained cells were then washed once with FACS buffer (2% FBS-PBS-0.02% NaN_3_).

A tube containing PBMCs (1 × 10^6^ cells) in 50 μL of 10% human AB serum was placed on ice for 30 min. The cells were stained with a cocktail of antibodies (Biolegend, San Diego, CA, USA) against CD3-APC/Cy7 (clone SK7), CD4-Alexa Fluor 488 (clone SK3), CD8-PE/Cy7 (clone SK1), CD45RA-APC (clone HI100), CCR7-PE (clone G043H7) and PD1-PE/Dazzle 594 (clone NAT105). The stained cells were placed on ice for 30 min before washing twice with cold FACS buffer, fixed with 300 μL of 1% paraformaldehyde in PBS, and then subjected to flow cytometry. Data were collected at 50,000 events using a BD LSRFortessa flow cytometer (BD Biosciences, San Jose, CA, USA) and analyzed with FlowJo software (BD Biosciences).

### 2.9. Statistical Analysis

Normally distributed continuous variables were summarized as means ± standard deviations; otherwise, medians (interquartile range, IQR) were used. Categorical variables are described as percentages and compared with the chi-square test. Continuous variables are described using geometric mean antibody titers (GMT), medians, and IQR. The Mann–Whitney U test was used to compare differences between groups. The generalized linear model was applied to evaluate the association between immune responses (including anti-RBD IgG and sVNT) and possible factors (vaccination, sex, and age). The statistical significance of anti-RBD IgG, sVNT, and other factors was determined using Kruskal–Wallis and Dunn’s multiple comparison tests using GraphPad Prism, version 9 (GraphPad Software Inc., San Diego, CA, USA) and STATA Statistical Software, release 17 (StataCorp LLC, College Station, TX, USA). Two-tailed *p* values less than 0.05 were considered significant (* *p* < 0.05, ** *p* < 0.01, *** *p* < 0.001, and **** *p* < 0.0001).

## 3. Results

### 3.1. Demographic Characteristics and Clinical Manifestations

The study enrolled 292 participants with prior COVID-19 infection who lived in Bangkok or its surrounding areas. Of these, 158 participants (86.1% women) were in the LC group and 134 participants (66.4% women) were in the HC group (Figure 1). The mean age of both groups was similar (Table 1). Of the participants in the HC and LC groups, 130 (97%) and 157 (99.4%), respectively, were treated in outpatient home isolation, indicating that their COVID-19 symptoms were mild to moderate. The median peak viral RNA (based on cycle threshold values) of the LC group (19.7 (interquartile range, IQR = 18.0–21.9)) was lower than that of the HC group (20.7 (IQR = 18.2–25.1), *p* < 0.037). The median of a targeted COVID-19 gene for the envelope (E) was 18.4 (IQR = 16.3–23.3) for the HC group and 17.8 (IQR = 16.5–19.9) for the LC group (*p* = 0.017).

Our results show no significant correlation between the vaccination status (either primary (2 doses) or booster shots) of the two groups (non-significant difference (ns)). Retrospective analysis revealed that patients in the LC group had a higher proportion of favipiravir treatment (65%) than those in the HC group (53.7%). This finding indicates that LC was associated with lower cycle thresholds and disease severity based on the prevalence of favipiravir treatment. Two comorbidities occurred more frequently in the LC group than in the HC group: hypertension (*p* = 0.052) and obesity (*p* = 0.025). Most participants had received more than three doses of vaccinations (93.0% in LC vs. 90.3% in HC, *p* = 0.306). The predominant residual LC symptoms were fatigue/myalgia (91.8%), dyspnea (74.7%), anorexia (55.1%), and problems with concentration (77.8%) and memory (63.9%). Other respiratory, gastrointestinal, and musculoskeletal symptoms were reported more frequently in the LC group than in the HC group (*p* < 0.05).

### 3.2. Immune Responses against SARS-CoV-2 Variants

High antibody titers were observed in both the LC and HC groups (Figure 2A). Anti-SARS-CoV-2 receptor-binding domain immunoglobulin G (anti-RBD IgG) positivity for SARS-CoV-2 S protein was observed in almost 100% of participants at all time points in both groups, except for one patient who had a negative IgG level at >84 days after a two-dose vaccination prior to breakthrough COVID-19 infection. Vaccination of at least three doses produced markedly high and comparable IgG levels in the LC and HC groups (ns), suggesting that the titer of anti-SARS-CoV-2 IgG antibody was still detectable at high levels in the two groups. Serum samples at different time points after illness onset showed differences in the distribution of IgG. The RBD-IgG geometric mean titers (GMT) were higher with the three-dose than the two-dose vaccinations at ≤84 days PC in both the LC and HC groups (*p* < 0.05). The overall proportion of sera with an IgG > 1000 binding antibody units per mL (BAU/mL) decreased from <56 days to 57–84 days after COVID-19 onset (PC, 86.8% to 79.6%) and again after 84 days PC (74.7%, ns for HC vs. LC, Figure 2A). This finding suggests that the recent booster vaccination effectively enhanced IgG responses and that IgG antibodies remained markedly high at 3 months PC. Moreover, the titers reached their peak at approximately 2 months PC and decreased slightly (approximately 25%) over the following 3 months (GMT 2713 to 2085 in the LC group and GMT 2482 to 2377 BAU/mL in the HC group). Sera from individuals presumably recently infected with BA.5/BA.4 (Figure 2A (black dots)) showed higher RBD-IgG GMT levels (4630 BAU/mL) than other groups (Figure 2A (pink and blue dots), *p* < 0.001). The findings imply that previous infection with BA.5/BA.4, but not ancestor Omicron variants (such as BA.1 or BA.2), offered a substantially stronger immune response regardless of the individual vaccine heterogeneity.

### 3.3. Neutralizing Antibody Responses

Ninety-nine percent of patients had positive surrogate virus neutralization (sVNT) against the wild-type and Omicron variants (Figure 2B). Consistent with previous studies [29] on the ability of Omicron to escape the immune system, particularly BA.4 and BA.5, both the early and late antibody responses were directed toward the ancestral SARS-CoV-2 strain. In the LC group, the sVNT level against Wuhan was high regardless of the number of vaccine doses and at all time points PC, even after more than 84 days PC. Positive sVNT titers were mostly observed < 56 days PC (Omicron GMT 59.3%), and sVNT against the Omicron variant was approximately 30% lower as compared with the Wuhan variant at most time points. The sVNT was more sustained for three-dose vaccinations than for ≤two-dose vaccinations. In the HC group, sVNT was also high against the Wuhan variant regardless of the number of vaccine doses and at all time points PC. Additionally, sVNT against the Omicron variant was 30% lower than that against the Wuhan variant. Of note, sVNT in the HC group peaked at 57–84 days PC compared with the LC group, in which sVNT remained at its highest level after 84 days.

We also compared the microneutralization activity against BA.1, BA.5 and Wuhan using an authentic virus neutralization assay (NeuT). We analyzed the NeuT of samples collected ≤ 56, 57–84, and >84 days after infection during the BA.1 and BA.2 period and samples collected at <29 days after infection during the BA.5/BA.4 era. For all NeuT tested in this study, the neutralizing ability declined remarkably at >84 days after infection. The HC and LC groups showed low neutralizing antibody titers against the Omicron variant BA.1 and BA.5 (Figure 2C). Samples from the LC group generally had 2–3-fold lower neutralizing activities than those from the HC group (Figure 2C). In the vaccinated groups, BA.5 NeuT was comparable to BA.2 NeuT, and this was independent of boosting status (Figure 2D). Moreover, boosting vaccination, with either 1 dose or 2 doses, elicited higher levels of antibody titers against the Wuhan virus in the LC group with a modest response against the Omicron variants, suggesting that the efficacy and importance of memory humoral immunity conferred by recent vaccination was high. However, in contrast with the LC group, the HC group showed higher NeuT against BA.5 and BA.1 at most time points (Figure 2C), but these NeuT were still significantly lower compared with those against Wuhan. Collectively, these data suggest that the Omicron BA.5/4 and BA.1 strains may lead to re-infection because of their distinct ability to evade previously acquired immunity. In addition, the neutralizing antibody titers against Wuhan and Omicron variants recapitulated the trend in which antibody responses are determined by sVNT.

### 3.4. Immune Response after a Booster Vaccine

The analysis investigated the immunogenicity of a subgroup of participants who received a booster dose of Pfizer-BioNTech mRNA vaccine 4 weeks after their initial vaccination. The results show that the levels of IgG antibodies to RBD doubled after booster vaccination (*p* < 0.0001), indicating an enhanced immune response against SARS-CoV-2. However, there was a slight difference in the mean GMT concentration of anti-RBD IgG between the low-risk (LC) and high-risk (HC) groups, with the LC group having a 10% higher concentration of anti-RBD IgG (4665 ± 1.76 vs. 4152 ± 2.15 BAU/mL, *p* < 0.05) (Figure 3A).

Interestingly, the study also found that the sVNT against the Omicron variant was similarly increased in almost all participants, regardless of whether they had LC or HC (LC 92.87% vs. HC 90.09%, ns (Figure 3B)). This suggests that mRNA booster vaccination may enhance immunity after COVID-19 infection, especially for vulnerable individuals. It is important to note that these results are specific to the Pfizer-BioNTech mRNA vaccine and may not apply to other COVID-19 vaccines or vaccine combinations. Additionally, the study only investigated the immune response 4 weeks after booster vaccination, and further research is needed to determine the duration of the enhanced immune response and its effectiveness against other SARS-CoV-2 variants.

### 3.5. Characterization and Distribution of Memory T Cell Subsets

To assess the correlation of immune responses to memory T cell subsets between HC and LC individuals, we analyzed the phenotype of CD4^+^ T cells, CD8^+^ T cells and memory T cell subsets between the time of symptom onset of COVID-19 using a multicolor flow cytometer. T cells were characterized using major phenotypic markers, including CD3, CD4, and CD8, and memory phenotypic markers, including CD45RA and CCR7. The memory CD4^+^ T cell and CD8^+^ T cell subsets were subcategorized into naïve (CCR7^+^CD45RA^+^), central memory (CM, CCR7^+^CD45RA^−^), effector memory (EM, CCR7^−^CD45RA^−^) and terminally differentiated effector memory (TEMRA, CCR7^−^CD45RA^+^) and programmed cell death protein 1 (PD1) expression. No difference was found in the four subsets of memory CD4^+^ T cells, including naïve, central memory, effector memory, and TEMRA, between the HC and LC groups (Figure 4).

We found that the percentage of CD8^+^ T cells was similar in the HC and LC groups (Figure 5B). Interestingly, significant differences in the percentage of CD8^+^ T cells in the LC group was demonstrated. The percentage of CD8^+^ T cells from LC with ≤56 days PC was significantly reduced compared with the group with 57–84 days PC (Figure 5B). Among the subsets of memory CD8^+^ T cells, we observed no significant difference in naïve, effector memory, and TEMRA cells between the HC and LC groups (Figure 5C,E,F). Interestingly, effector memory CD8^+^ T cells were higher in the LC group after >84 days PC than in the LC group after 57–84 days PC (Figure 5E). We found significant differences in central memory CD8^+^ T cells between the HC and LC groups. The LC group at ≤56 days and 57–84 days PC had lower expressions of the percentage of central memory CD8^+^ T cells than the HC groups after ≤56 days and 57–84 days PC, respectively (Figure 5D).

Expression of PD1 was determined in each subset of memory CD4^+^ and memory CD8^+^ T cells. We found significant differences in the expression of PD1 in central memory and effector memory CD4^+^ T cells between the LC and HC groups. The LC groups with illness onset at ≤56 days, 57–84 days and >84 days had lower percentages of PD1 positive cells than the HC groups (Figure 6B,C). However, there were no significant differences in PD1 expression in TEMRA CD4^+^ T cells between the two groups (Figure 6D). Similar to CD4^+^ memory T cells, significant differences in PD1 expression in central CD8^+^ memory T cells were demonstrated between the LC and HC groups (Figure 6E). The expression of PD1 in the LC groups with ≤56 days, 57–84 days and >84 days PC was lower than that in the HC groups. However, there were no significant differences in PD1 expression in effector memory and TEMRA CD8^+^ T cells between the LC and HC groups (Figure 6F,G).

## 4. Discussion

In this study, we identified the following key findings. LC shows a constellation of enervating symptoms that most commonly includes incessant fatigue/myalgia, post-exertional malaise, shortness of breath, anorexia, and cognitive dysfunction [30]. Consistent with previous studies, the current investigation found that this syndrome was prevalent in patients with mild-to-moderate COVID-19, in women, and in middle-aged patients regardless of vaccination status [10,31]. Age is considered a common determinant of disease severity, primarily because people’s immune responses progressively deteriorate with age [32]. Given the uncertainty about the frequency and prevalence of individual symptoms of LC and their duration, there are patient reports of fatigue beginning shortly after COVID-19 recovery. A systematic review of self-reported fatigue after recovery from COVID-19 infection revealed that fatigue could persist for up to 6 months and that immunologic dysfunction can last up to 8 months after mild-to-moderate COVID-19 [19,29]. SARS-CoV-2 can persist in some patients with long-term symptoms, leading to chronic inflammation and some organ and tissue dysfunction [33]. In a previous cohort study, the chronic “brain fog” attributed to COVID was significantly associated with gender (female), respiratory symptoms at disease onset, and disease severity [34]. However, there is increasing evidence that LC may develop independently of the severity of initial symptoms [35]. The development of LC may depend on the persistence of the antigen and a sustained specific immune response to SARS-CoV-2 [36].

Second, a detailed understanding of innate and adaptive immune responses to LC in conjunction with vaccination status and heterogeneous characteristics of viral and host responses is essential for planning therapeutic and vaccination strategies. Our results show no significant differences in immune responses between individuals with LC and HCs, regardless of vaccination status before COVID-19 infection. Significant decreases in IgG and neutralizing antibodies were also observed in participants without booster vaccines prior to COVID-19 infection. Such decreases are classically associated with waning immunity. Markedly high levels of antibodies were still observed at 3 months PC. Analysis of the immune responses of individuals presumed infected with BA.5/BA.4 revealed a significant increase in immunity compared with individuals infected with other Omicron variants. More interestingly, in a subset of participants with LC, anti-RBD IgG antibodies, but not sVNT, were significantly increased after Pfizer-BioNTech booster shot compared with the HC group. These findings are supported by prior reports that LC pathogenesis might include persistent viral antigen, reactivation of latent herpesviruses, and chronic inflammation, which could be associated with the increase in antibody responses [12,37].

T cells and memory T subsets have been studied and considered responsible for severity and recovery in COVID-19 patients. Several studies have reported that numbers of CD3^+^, CD4^+^ and CD8^+^ T cells were reduced in severe COVID-19 patients compared with non-severe or healthy patients [38,39,40]. In this study, CD4^+^ T cells and CD8^+^ T cells show no differences in either HC or LC patients.

Memory T cells are an important and diverse subset of antigen experienced T cells that are maintained over the long term and converted into effector cells upon re-exposure when needed. Previous studies have reported that naïve T cells decreased in convalescent COVID-19, but effector and central memory subsets increased proportionally and remain in the circulation for several months [41]. However, most of these studies were conducted during acute/short periods of COVID-19 infection. In a recent study, the researcher examined long COVID-19 patients and discovered that the number of naïve CD4^+^ T cells was similar in long COVID-19 and healthy, recovered individuals. The number of central memory CD4^+^ T cells was reduced in long recovering severe patients. Naïve and central memory CD8^+^ T cells were decreased in LC patients [42]. In this study, we analyzed the distribution of memory T cells subsets in HC and LC. We found that naïve CD4^+^ T cells and naïve CD8^+^ T cells were similar in both groups. The central memory of CD4^+^ T cells, effector memory of CD4^+^ T cells and effector memory of CD8^+^ T cells were also similar in both groups. Similar to a previous study, we found that the central memory of CD8^+^ T cells was not effectively activated in the LC patients in the first few months after COVID-19 infection. This implies that there are reduced CD8^+^ T cell activities against SARS-CoV-2 in LC patients.

TEMRA cells are the effector memory T cells that re-express CD45RA (a marker found on naïve T cells). TEMRA are terminally differentiated cells that display the shortest telomeres. They carry higher levels of cytotoxic and exhaust genes compared with EM T cells. TEMRA cells express higher levels of inhibitory molecules, including CD57, killer lectin inhibitory receptor 1(KLRG1) and programed cell death protein 1 (PD1). They produce effector cytokines, mainly IFN-γ and TNF-α, but have low proliferative capacity and are short lived. It was found that high levels of virus-specific TEMRA cells were maintained after dengue vaccination [43] and that the increase in TEMRA cells persisted for 6–7 months in COVID-19 patients [44]. CD8^+^ TEMRA cells have mostly been studied. CD8^+^ TEMRA cells were found to be highly increased in chronic viral infections (e.g., CMV, HIV and HCV), autoimmune disorders and cancers. The development of CD8^+^ TEMRA depends on the high inflammatory milieu and high antigen dose [45,46]. CD8^+^ TEMRA cells were increased in hospitalized COVID-19 patients and persisted for 6 weeks. In our study, CD8^+^ TEMRA cells were not different in HC and LC patients. Functions of CD4^+^ TEMRA cells are still elusive. It was found that the population of CD4^+^ TEMRA cells was not altered in COVID-19 patients [44]. In our study, the CD4^+^ TEMRA cells were similar in HC and LC patients.

In COVID-19 patients, PD1 expression was found to increase in T cell exhaustion subsets and to play a role in post-COVID-19 immunity dysfunctions [47]. TEMRA that expressed PD1 were characterized as T cell exhaustion subsets. Several studies have reported that exhaustion of T cells is related to the severity of COVID-19 infection, except in cases of mild and moderate convalescence [42]. Similar to our study, in which most cases had mild-to-moderate symptoms, PD1 expression in TEMRA CD4^+^ and CD8^+^ T cells did not differ in HC and LC patients. Though it is known that the expression of inhibitory receptors, including PD1 in subsets of T cell, are attributed to T cell exhaustion, Min-Seok Rha et al. found that PD1 expressing SARS-COV-2 specific CD8 T cells are not exhausted but remain active in COVID-19 patients. Their findings are more likely to indicate that PD1 expression indicates the early activation of T cells [24,48]. In this study, we found that PD1 expression in central memory CD4^+^ T cells, effector memory CD4^+^ T cells, and central memory CD8^+^ T cells was lower in the LC groups than in the HC groups. This finding may imply that PD1 expression in central memory and effector memory T cells in HC groups likely remained functional rather than exhausted. Previous studies have shown that cases of mild and severe COVID-19 exhibit similar post-COVID-19 symptoms and that different immune endotypes correlated with different post-COVID-19 symptoms. In contrast with severe COVID-19 symptoms, dysregulation of immunity CD4, CD8 T cells and regulatory T cells is exclusively correlated with the severity of COVID-19 infection [12,49]. This refers to the different mechanisms and immune cell subsets that might be responsible for post-COVID-19 symptoms.

This study had some limitations. During the study period, most COVID-19 cases had mild-to-moderate symptoms, and those affected were most likely to be able to treat themselves at home. Consequently, we had only patients under home isolation conditions. This means that fair comparisons with LC may not be possible in hospitalized patients, and the results cannot be generalized to the broader LC community. In addition, our study focused on post-acute residual COVID-19 symptoms at 1–4 months PC. Therefore, the study could not determine who was likely to develop LC. Another limitation is the sample size. Our patients with classic LC symptoms and risk factors may represent only a tiny subset, making it difficult to identify specific, novel predictors. In addition, it has been well established that inflammatory cytokines, such as IL-6, IL-8, IL-1β, TNF-α, IFNγ-induced protein 10 (IP-10), and granulocyte-macrophage colony-stimulating factor (GM-CSF), and chemokines such as CC motif ligand 2 (CCL2), CCL-5, and CCL3, are typically produced by macrophages, mast cells, endothelial cells, and epithelial cells during the innate immune response [50]. Our unpublished data indicate elevated IL-6 levels in some COVID-19 patients. However, we need further clarification on the factors that predispose individuals to cytokine storms and other inflammatory cytokine reactions. Therefore, the study did not encompass an investigation into cytokines. Lastly, the study had a higher representation of female participants. Numerous research studies [51,52] have indicated that females exhibit more symptoms than males, not only during the initial phase but also in the subsequent follow-up period. Males face an elevated likelihood of hospitalization and ICU admission, suggesting a greater potential for severe acute illness. Conversely, females had a heightened susceptibility to experiencing a heavier load of self-reported acute and persistent symptoms. These results emphasize the necessity for additional research and reporting on the gender-based aspects of COVID-19 disease.

## 5. Conclusions

In summary, our study provides immunological evidence for vaccination programs after COVID-19 infection with or without LC symptoms. We found that an up-to-date vaccine booster dose PC resulted in higher anti-spike IgG and sVNT. Furthermore, we examined the immune escape capacity of the BA.1 and BA.5/BA.4 Omicron sub-lineages after COVID-19 infection. Our data suggest that, in the case of Omicron infection, previous vaccination promotes a marginally stronger humoral immune response with a broader spectrum of neutralization against the ancestral strain and the Omicron sub-lineages. All participants responded well to an mRNA vaccine booster dose following COVID-19 infection. Sera from participants infected during the Omicron BA.5/BA.4 wave, but not during the Omicron BA.1 or BA.2 wave, showed better immune responses against all Omicron variants. In LC central memory CD4^+^ T cells, effector memory CD4^+^ T cells, and central memory CD8^+^ T cells were reduced in the first months PC; therefore, booster vaccines may be needed to rescue T cell function for the LC group as compared with the HC group. As Omicron continues to evolve, this poses great challenges to currently available immunotherapies and vaccines.

## Figures and Tables

**Figure 1 vaccines-11-01413-f001:**
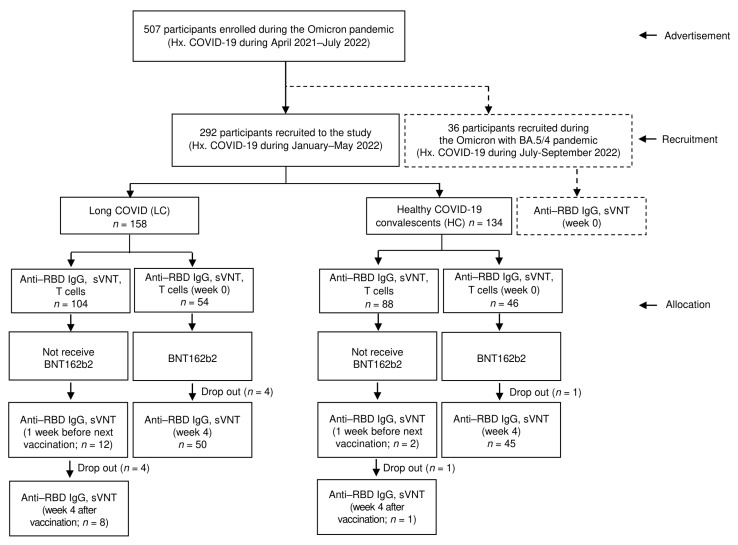
Recruitment and enrolment for the study.

**Figure 2 vaccines-11-01413-f002:**
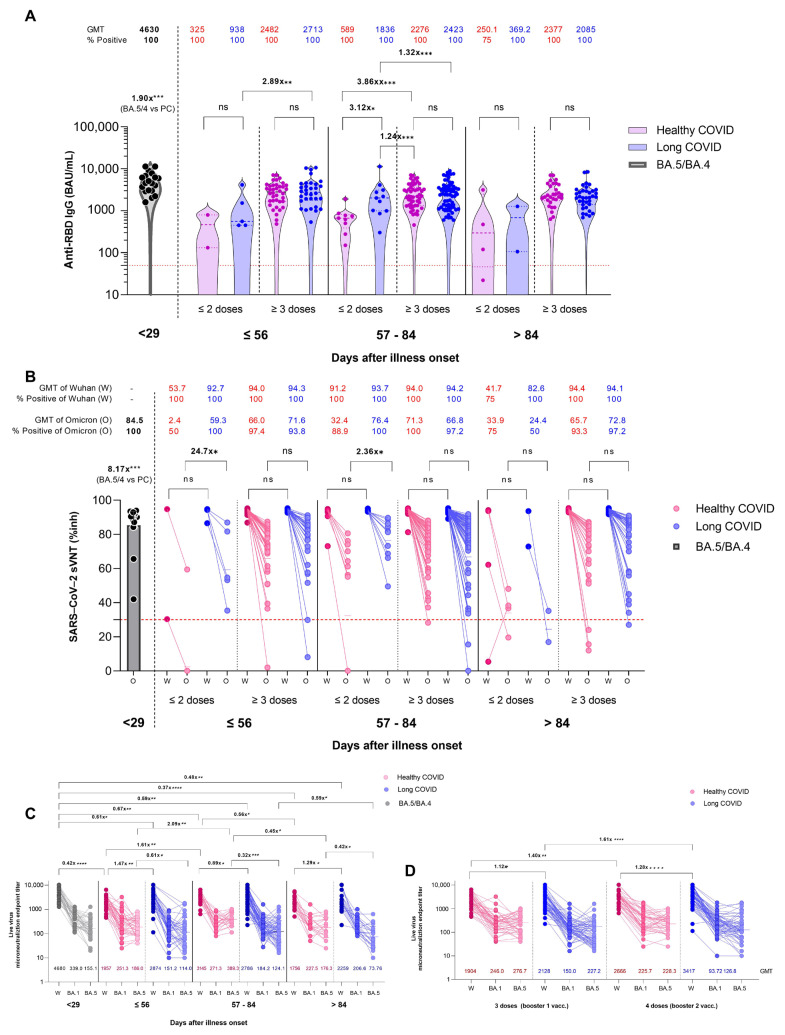
Comparison of immune responses of patients following breakthrough COVID-19 infection with prior vaccination during the Omicron pandemic. (1) Patients with healthy COVID recovery (HC, red dots), (2) patients with residual long COVID (LC, blue dots) and (3) patients during the BA.5/BA.4 wave (black color). (**A**) Geometric mean titers (GMTs) of SARS-CoV-2 anti-spike protein receptor-binding domain antibodies (anti-RBD IgG) in serum samples obtained from patients after COVID-19 infection and with previous varying vaccination status and duration PC. All sera were obtained from patients during the Omicron pandemic. The dotted line (red) represents the threshold for positive assay results. (**B**) Scatter plots illustrating inhibition rates of Wuhan and Omicron RBD-blocking antibodies, measured using a surrogate virus neutralization test (sVNT) by vaccination/reinfection status. The lower dotted line represents the cutoff for seropositivity. (**C**) Microneutralization endpoint titers against Wuhan and Omicron sub-lineages BA.2 and BA.5 live viruses in sera at different time points PC between HC, LC and BA.5/BA.4 groups. (**D**) Microneutralization endpoint titers against Wuhan and Omicron sub-lineages BA.2 and BA.5 live viruses in sera from individuals who had received 3 or 4 doses of vaccine between HC and LC groups. The generalized linear model was applied to evaluate the associations between the immune response (including IgG antibody and sVNT responses) and possible factors (vaccination, sex, and age). * *p* < 0.05; ** *p* < 0.01; *** *p* < 0.001; **** *p* < 0.0001. ns is a non-significant difference.

**Figure 3 vaccines-11-01413-f003:**
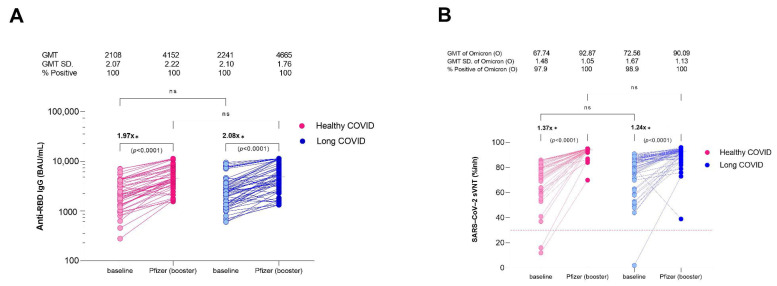
Comparison of immunogenicity against SARS-CoV-2 in patients with healthy COVID recovery (HC, red color) and patients with residual long COVID (LC, blue color) after a booster dose of the Pfizer-BioNTech mRNA vaccine, by scatter plot analysis, during the Omicron pandemic. (**A**) SARS-CoV-2 anti-spike protein antibodies to the receptor-binding domain (anti-RBD IgG) in serum samples from subjects 4 weeks after a 30 μg booster dose of Pfizer vaccine. (**B**) The inhibition rate of Omicron RBD antibodies was measured using a surrogate virus neutralization test (sVNT). The generalized linear model was applied to evaluate the association between the immunogenicity of anti-RBD IgG and the sVNT, and to identify potential predictive factors (sex and age). * *p* < 0.0001. ns is a nonsignificant difference.

**Figure 4 vaccines-11-01413-f004:**
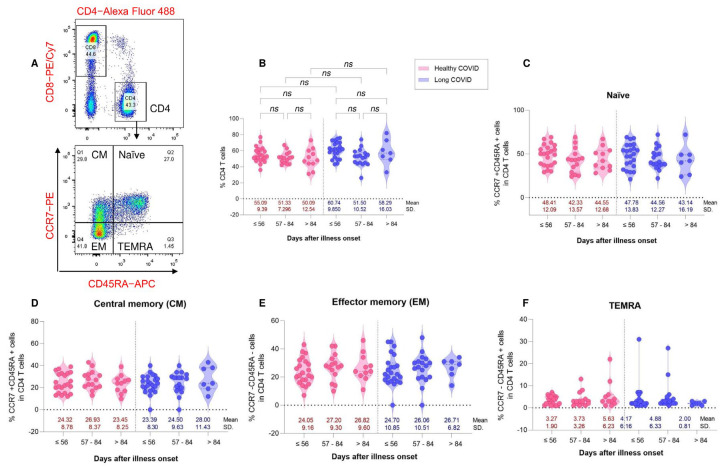
Distribution of CD4^+^ T cells and CD4^+^ memory T cell subsets from healthy COVID recovery (HC, red color) and residual long COVID (LC, blue color). PBMCs were collected from HC (N = 48) and LC (N = 48) patients. PBMCs were stained with antibodies for multi-color flow cytometry to analyze the immunophenotype of T cell subsets. (Upper (**A**)) CD3^+^ cells were gated and CD4 and CD8 were plotted and the representative flow plot of naïve and memory CD4^+^ T cell subsets (lower (**A**)). (**B**) Percentages of CD4^+^ T cells in each subgroup of HC and LC patients. Respective percentages of naïve (CCR7^+^CD45RA^+^), central memory (CM, CCR7^+^CD45RA^−^), effector memory (EM, CCR7^−^CD45RA^−^) and terminally differentiated effector memory (TEMRA, CCR7^−^CD45RA^+^) subsets in CD4^+^ T cells (**C**–**F**). Data are represented as mean with 95% CI. ns is a non-significant difference.

**Figure 5 vaccines-11-01413-f005:**
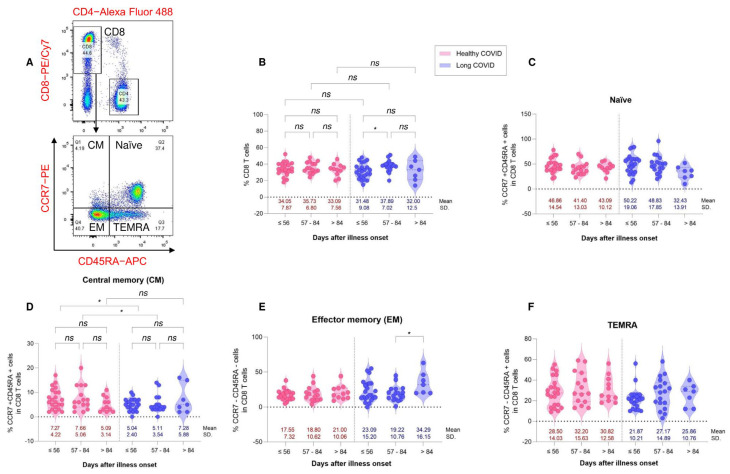
Distribution of CD8^+^ T cells and CD8^+^ memory T cell subsets of healthy COVID recovery (HC, red color) and residual long COVID (LC, blue color). PBMCs were collected from HC (N = 48) and LC (N = 48) patients. PBMCs were stained with antibodies for multi-color flow cytometry to analyze the immunophenotype of T cell subsets. (Upper (**A**)) CD3^+^ cells were gated and CD4 and CD8 were plotted. (Lower (**A**)) The representative flow plot of naïve and memory CD8^+^ T cell subsets. (**B**) Percentages of CD8^+^ T cells in each subgroup of HC and LC patients. Respective percentages of naïve (CCR7^+^CD45RA^+^), central memory (CM, CCR7^+^CD45RA^−^), effector memory (EM, CCR7^−^CD45RA^−^) and terminally differentiated effector memory (TEMRA, CCR7^−^CD45RA^+^) subsets in CD4^+^ T cells (**C**–**F**). Data are represented as mean with 95% CI, with significance of * *p* ≤ 0.05. ns is a non-significant difference.

**Figure 6 vaccines-11-01413-f006:**
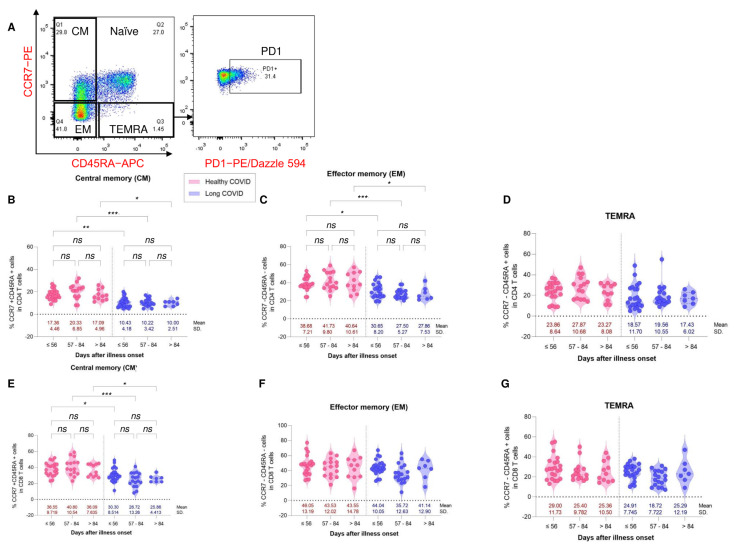
Expression of PD1 in memory T cell subsets from healthy COVID recovery (HC, red color) and residual long COVID (LC, blue color). PBMCs were collected from HC (N = 48) and LC (N = 48) patients. PBMCs were stained with antibodies for multi-color flow cytometry to analyze the immunophenotype of T cell subsets. CD3^+^ CD4^+^ cells and CD3^+^ CD8^+^ cells were gated. (Left (**A**)) CCR7 and CD45RA were plotted. (Right (**A**)) Representative flow plot of PD1 expressing cells was plotted. (**B**–**D**) Respective percentages of PD1^+^ cells in central memory (CM, CCR7^+^CD45RA^−^), effector memory (EM, CCR7^−^CD45RA^−^) and terminally differentiated effector memory (TEMRA, CCR7^−^CD45RA^+^) subsets in CD4^+^ T cells. (**E**–**G**) Respective percentages of PD1^+^ cells in central memory (CM, CCR7^+^CD45RA^−^), effector memory (EM, CCR7^−^CD45RA^−^) and terminally differentiated effector memory (TEMRA, CCR7^−^CD45RA^+^) subsets in CD8^+^ T cells. Data are represented as mean with 95% CI, with significance of * *p* ≤ 0.05, ** *p* ≤ 0.001, *** *p* ≤ 0.0001. ns is a non-significant difference.

**Table 1 vaccines-11-01413-t001:** Demographic and clinical characteristics of participants *.

Characteristics			Healthy COVID-19 Convalescents (HC) Group (*n* = 134)		Long COVID (LC) Group (*n* = 158)							*p*-Value ^‡^
			*n*	(%)	*n*	(%)	Long COVID Symptoms				*p*-Value ^†^	
							3 Major Symptoms (*n* = 132)		1–2 Major Symptoms (*n* = 26)			
							*n*	(%)	*n*	(%)		
Male/Female			45/89	33.6/66.4	22/136	13.9/86.1	18/114	13.6/86.4	4/22	15.4/84.6	0.814	<0.001
Age (yr.), mean (SD)			36.3	(9.6)	38.4	(8.4)	38.7	(8.5)	37.0	(7.5)	0.338	0.078
	18–30		42	31.3	24	15.2	19	14.4	5	19.2	0.344	0.362
	31–45		72	53.7	99	62.7	81	61.4	18	69.2		
	>45		20	14.9	35	22.2	32	24.2	3	11.5		
Time from symptom onset to COVID detection, months												0.353
	1–2		40	29.9	37	23.4	34	25.8	3	11.5	0.147	
	3		60	44.8	83	52.5	65	49.2	18	69.2		
	>3		34	25.4	38	24.1	33	25.0	5	19.2		
Types of hospital admission											0.024	0.123
	Home isolation		130	97.0	157	99.4	132	100.0	25	96.2		
	Hospitel		4	3.0	1	0.6	0	0.00	1	3.8		
COVID-19 detection											0.379	0.362
	RT-PCR ^§^		129	96.3	146	92.4	122	92.4	24	92.3		
	ATK ^¶^		5	3.7	12	7.6	10	7.6	2	7.7		
Cycle threshold **, median (IQR)			20.7	(18.2–25.1)	19.7	(18.0–21.9)	19.7	(17.7–22.1)	19.5	(18.5–21.0)	0.921	0.039
	<20		52	43.7	80	55.9	64	54.2	16	64.0	0.466	0.037
	20–30		51	42.9	55	38.5	48	40.7	7	28.0		
	>30		16	13.4	8	5.6	6	5.1	2	8.0		
Envelope, median (IQR)			18.4	(16.3–23.3)	17.8	(16.5–19.9)	17.7	(16.3–19.7)	18.1	(17.0–19.9)	0.588	0.017
RNA-dependent RNA polymerase (RdRp), median (IQR)			19.9	(18.0–24.8)	19.2	(17.9–22.5)	19.4	(17.8–21.4)	18.9	(18.1–21.5)	0.670	0.807
Presence of comorbidities												
	None		121	90.3	125	79.1	104	78.8	21	80.8	0.820	0.009
	Hypertension		5	3.7	15	9.5	10	7.6	5	19.2	0.064	0.052
	Obesity		3	2.2	13	8.2	12	9.1	1	3.8	0.374	0.025
	Diabetes mellitus		5	3.7	6	3.8	5	3.8	1	3.8	0.989	0.976
	Cancer		0	0.0	3	1.9	3	2.3	0	0.0	0.349	0.109
	Chronic respiratory disease		2	1.5	3	1.9	3	2.3	0	0.0	0.438	0.791
	Kidney disease		0	0.0	1	0.6	1	0.8	0	0.0	0.656	0.356
Lifestyle												
	Smoking										0.027	0.016
		Never	114	85.1	144	91.1	122	92.4	22	84.6		
		Stop smoking	12	9.0	5	3.2	2	1.5	3	11.5		
		Smoking	8	6.0	9	5.7	8	6.1	1	3.9		
	Vaccination										0.485	0.306
		0–2 doses	13	9.7	11	7.0	9	6.8	2	7.7		
		1 booster dose	66	49.3	64	40.5	52	39.4	12	46.2		
		2 booster doses	55	41.0	83	52.5	71	53.8	12	46.2		
	Medication										0.922	0.048
		No treatment	9	6.7	3	1.9	3	2.3	0	0.0		
		Symptomatic treatment	53	39.6	51	32.5	42	32.1	9	34.6		
		Symptomatic + Favipiravir treatment	69	51.5	94	59.8	79	60.3	15	57.7		
		Symptomatic + Favipiravir treatment + Dexamethasone	3	2.2	9	5.7	7	5.3	2	7.7		
Long COVID symptoms												
	Fatigue/myalgia		0	0.0	145	91.8	126	95.5	19	73.1	<0.001	<0.001
	Breathlessness		0	0.0	118	74.7	116	87.9	2	7.7	<0.001	<0.001
	Anorexia		0	0.0	87	55.1	85	64.4	2	7.7	<0.001	<0.001
	Problem with concentration		0	0.0	123	77.8	116	87.9	7	26.9	<0.001	<0.001
	Problem with memory		0	0.0	101	63.9	99	75.0	2	7.7	<0.001	<0.001
	Headache		40	29.8	88	55.7	75	56.8	13	50.0	0.522	<0.001
	Cough/chest pain/chest discomfort		52	38.8	111	70.3	92	69.7	19	73.1	0.730	<0.001
	Loss of smell/taste		23	17.2	52	32.9	43	32.6	9	34.6	0.840	0.001
	Muscle pain/joint pain		37	27.6	91	57.6	78	59.1	13	50.0	0.391	<0.001
	Diarrhea		20	14.9	46	29.1	39	29.5	7	26.9	0.788	0.004

* Continuous data demographic, clinic finding of all patients referred back presented as mean (standard deviation; SD), median (interquartile rang; IQR), and range at *p* < 0.05 indicates statistical significance. ^†^ The statistical significance was assessed by the Fisher’s exact test and Kruskal–Wallis test, statistical difference within long COVID symptoms at *p* < 0.05. ^‡^ The statistical significance was assessed by the Fisher’s exact test and Kruskal–Wallis test, statistical difference between participants with healthy COVID-19 convalescents (HC) and Long COVID (LC) group at *p* < 0.05. **^§^** RT-PCR is a real-time reverse transcription polymerase chain reaction (rRT-PCR) test for the qualitative detection of nucleic acid from SARS-CoV-2 in upper and lower respiratory specimens. **^¶^** ATK is SARS-CoV-2 rapid antigen self-test kits. ** Cycle threshold (Ct) value from RT-PCR tests represents the cycle number at which the signal breaches the threshold for positivity, a lower Ct value is indicative of a high viral load.

## Data Availability

Raw data used in this study, including de-identified patient metadata and test results, are available upon request.

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
