# Peer review of "Immune Response after SARS-CoV-2 Infection with Residual Post-COVID Symptoms"

_vaccines, 2023, doi:10.3390/vaccines11091413_

Round 1
Reviewer 1 Report
The authors provide a very comprehensive study on cellular and humoral immune responses in patients with SARS CoV2 infection with long covid compared with those who recovered well from the infection. Kinetics of vaccination responses is also investigated.
Comments:
Line 57. variants1, should read variants[1]
Line 155. previously45, should read previously [45]
Line 185. 'The 184 diluted blood samples produced a good mononuclear-cell purity.' How was this assessed? More detail is required here.
Lines 211- 217. There is insufficient detail regarding the flow cytometric immunophenotyping. What was the source of the antibodies? Which clones were use? Which fluorochromes were used? How many events were collected?
Line 255 -table 1. This is a very large table and, possibly due to formatting issues, is difficult to understand. The positioning of the 'healthy covid' and 'long covid' titles is not quite right. It also looks like the data is only applicable to females?
Line 366 - 369. 'CD4+ T cell and CD8+ T cell subsets were subcategorized into naïve (CCR7+CD45RA+), central memory (CM, CCR7+CD45RA-), effector memory (EM, CCR7+CD45RA-) and terminally differentiated effector memory (TEMRA, CCR7-CD45RA+) and programmed cell death protein 1 (PD1) expression.' The phenotype for CM and EM cells appears to be the same and is not consistent with the data/figure legends shown in figures 4 and 5.
Figures 4 and 5 - Please include the fluorochrome conjugate for the antibodies used on the flow cytometry dot plots.
The manuscript is well written and contains a lot of complex data. Some minor typos are evident.
Author Response
Response to Reviewer 1’s Comments
The authors provide a very comprehensive study on cellular and humoral immune responses in patients with SARS CoV2 infection with long covid compared with those who recovered well from the infection. Kinetics of vaccination responses is also investigated.
Line 57. variants1, should read variants [1]
Line 155. previously45, should read previously [45]
Responses 1: We thank for your valuable comments. We have addressed all minor changes in Line 57, 155.
Point 2: Line 185. 'The 184 diluted blood samples produced a good mononuclear-cell purity.' How was this assessed? More detail is required here.
Responses 2: The method for isolation of PBMCs was added in Line 180-183.
Lines 180-183: Peripheral blood was drawn into a tube supplemented with sodium heparin as anticoagulant [28]. The blood sample was diluted 1:1 with phosphate-buffered saline, pH 7.2 (PBS; HyClone, Logan, UT, USA) and PBMCs were isolated by using Ficoll-gradient centrifugation.
Point 3: Line 255 -table 1. This is a very large table and, possibly due to formatting issues, is difficult to understand. The positioning of the 'healthy covid' and 'long covid' titles is not quite right. It also looks like the data is only applicable to females.
Response 3: For table 1. We have inserted male/female and their ratios to make this clearer. In terms of the distribution of male vs. female participants. It was true that females were dominant in the study unavoidably. We inserted as the following information in Discussion Lines: 534-541
“Lastly, the study had a higher representation of female participants. Numerous research studies [49,50] have indicated that females exhibited more symptoms than males, not only during the initial phase but also in the subsequent follow-up period. Males faced an elevated likelihood of hospitalization and ICU admission, suggesting a greater potential for severe acute illness. Conversely, females had a heightened susceptibility to experiencing a heavier load of self-reported acute and persistent symptoms. As a result, this emphasizes the necessity for additional research and reporting focusing on the gender-based aspects of COVID-19 disease.”
Point 4: Lines 211- 217. There is insufficient detail regarding the flow cytometric immunophenotyping. What was the source of the antibodies? Which clones were use? Which fluorochromes were used? How many events were collected?
Response 4: The details of the flow cytometric staining and fluorochromes, source of antibodies, clones of antibodies and collected events have been added to the revised manuscript (lines 209-217).
Lines 209-217: A tube containing PBMCs (1 × 106 cells) in 50 µL of 10% human AB serum was placed on ice for 30 minutes. The cells were stained with a cocktail of antibodies (Biolegend, USA) against CD3-APC/Cy7 (clone SK7), CD4-Alexa Fluor 488 (clone SK3), CD8-PE/Cy7 (clone SK1), CD45RA-APC (clone HI100), CCR7-PE (clone G043H7) and PD1-PE/Dazzle 594 (clone NAT105). The stained cells were placed on ice for 30 minutes before washing twice with cold FACS buffer, fixed with 300 µL of 1% paraformaldehyde in PBS, and then subjected to flow cytometry. Data were collected at 50,000 events using a BD LSRFortessa flow cytometer (BD Biosciences, San Jose, CA, USA) and analyzed with FlowJo software (BD Biosciences).
Point 5: Line 366 - 369. 'CD4+ T cell and CD8+ T cell subsets were subcategorized into naïve (CCR7+CD45RA+), central memory (CM, CCR7+CD45RA-), effector memory (EM, CCR7+CD45RA-) and terminally differentiated effector memory (TEMRA, CCR7-CD45RA+) and programmed cell death protein 1 (PD1) expression.' The phenotype for CM and EM cells appears to be the same and is not consistent with the data/figure legenfds shown in figures 4 and 5.
Figures 4 and 5 - Please include the fluorochrome conjugate for the antibodies used on the flow cytometry dot plots.
Response 5: Line 368: The “effector memory (EM, CCR7+CD45RA-)” was changed to “effector memory (EM, CCR7-CD45RA-)”. We are sorry for the error.
The fluorochromes were added in dot plots in Figures 4, 5 and 6
Reviewer 2 Report
The manuscript entitled ''Immune response after SARS-CoV-2 infection with residual 2 post-COVID symptoms'' presents a nice piece of work requesting for booster vaccine as evident by low memory and Inactivation markers such as PD-1. However, I have few questions from the authors;
1. The authors performed IgG RBD ELISA. Did they obtained result for the IgA titers in the blood from HC and LC individuals.
2. The authors performed RT-qPCR data but I am unable to locate the data.
3. The biggest concern is why the authors have performed Surrogate virus neutralization assay? Why the live-virus neutralization not performed?
4. I am very impressed with flow cytometry data, Please mention TEMRA as Terminally Differentiated Effectors as many reader are not from core immunology background.
5. Why did the authors not performed ICS like IFN-g or TNF-a assays?
6. Did the authors used another inactivation markers other than PD-1?
7. Did the authors collected nasal swabs from the individuals and tested for IgA and virus neutralization with swab?
8. Did the authors performed neutralization with new SARS-CoV-2 Omicron variants like XBB.1?
Author Response
Reviewer 2
The manuscript entitled ''Immune response after SARS-CoV-2 infection with residual 2 post-COVID symptoms'' presents a nice piece of work requesting for booster vaccine as evident by low memory and Inactivation markers such as PD-1. However, I have few questions from the authors;
Point 1. The authors performed IgG RBD ELISA. Did they obtain result for the IgA titers in the blood from HC and LC individuals.
Response 1: Thank you for your value question. As indicated by numerous reports, the virus-specific antibody responses encompass IgG, IgM, and IgA. The early humoral responses specific to SARS-CoV-2 were primarily characterized by IgA antibodies. Due to limitations inherent in serological surveys for assessing the prevalence of SARS-CoV-2 infection within specific groups, we solely rely on the persistence of IgG antibodies to identify individuals who have been infected.
Point 2. The authors performed RT-qPCR data but I am unable to locate the data.
Response 2: Thank you very much for your question. We have summarized this result in the Table 1 by reporting the cycle threshold of the nucleocapsid (N), envelope (E) of Sarbecovirus, and RNA-dependent RNA polymerase (RdRp).
Point 3. The biggest concern is why the authors have performed Surrogate virus neutralization assay? Why the live-virus neutralization not performed?
Response 3: We extend our gratitude for these comments. In fact, we conducted both surrogate virus neutralization tests (sVNT) (Lines: 158-165) and live-virus neutralization (NeuT) tests (Line 166-178, 295-313). In this study, we chose the sVNT as our primary approach due to its compatibility with BSL-2 containment, which streamlined the assessment of the majority of serum samples. On the other hand, the live-virus microneutralization assay necessitates BSL-3 containment, which restricts the number of samples that can undergo evaluation using live viruses. Additionally, during the study's duration, the laboratory lacked access to the live BA.1 and BA.2 virus variants, leading to the exclusion of live-virus microneutralization assays involving these specific variants.
Point 4. I am very impressed with flow cytometry data, please mention TEMRA as Terminally Differentiated Effectors as many reader are not from core immunology background.
Response 4 : Thank you very much for our valuable comments. We have added more details in Discussion Lines 481-497.
TEMRA cells are the effector memory T cells that re-express CD45RA (a marker found on naïve T cells). TEMRA are terminally differentiated cells that display the shortest telomeres. They carry higher levels of cytotoxic and exhaust genes compared to EM T cells. TEMRA cells express higher levels of inhibitory molecules including CD57, killer lectin inhibitory receptor 1(KLRG1) and programed cell death protein 1 (PD1). They produce effector cytokines mainly IFN-γ and TNF-α, but they have low proliferative capacity and are short lived. It was found that high levels of virus specific TEMRA cells were maintained after dengue vaccination [44] and that the increase in TEMRA cells persisted for 6-7 months in COVID-19 patients [43]. CD8+ TEMRA cells have mostly been studied. CD8+ TEMRA cells were found to be highly increased in chronic viral infections (e.g., CMV, HIV and HCV), autoimmune disorders and cancers. The development of CD8+ TEMRA depends on the high inflammatory milieu and high antigen dose (Anis L and Tamas F, 2013; Ahto Salmuets, et al., 2022). CD8+ TEMRA cells were increased in hospitalized COVID-19 patients and persisted for 6 weeks. In our study, CD8+ TEMRA cells were not different in HC and LC patients. Functions of CD4+ TEMRA cells are still elusive. It was found that the population of CD4+ TEMRA cells was not altered in COVID-19 patients [43]. In our study, the CD4+ TEMRA cells were similar in HC and LC patients.
References:
- Larbi, A.; Fulop, T. From "truly naïve" to "exhausted senescent" T cells: when markers predict functionality. Cytometry A 2014, 85, 25-35. doi: 10.1002/cyto.a.22351.
- Salumets, A.; Tserel, L.; Rumm, A. P.; Türk, L.; Kingo, K.; Saks, K.; Oras, A.; Uibo, R.; Tamm, R.; Peterson, H.; et al. Epigenetic quantification of immunosenescent CD8(+) TEMRA cells in human blood. Aging Cell 2022, 21, e13607. doi: 10.1111/acel.13607.
Point 5. Why did the authors not performed ICS like IFN-g or TNF-a assays?
Response 5 Thank you for inquiring about cytokines. We have added the following details in Line 521-530.
“It's well-established that inflammatory cytokines, such as IL-6, IL-8, IL-1β, TNF-α, IFNγ-induced protein 10 (IP-10), granulocyte-macrophage colony-stimulating factor (GM-CSF), and chemokines like CC motif ligand 2 (CCL2), CCL-5, and CCL3, are typically produced by macrophages, mast cells, endothelial cells, and epithelial cells during the innate immune response [48]. Our unpublished data indicate elevated IL-6 levels in some COVID-19 patients. However, we need further clarification on the factors that predispose individuals to cytokine storms and other inflammatory cytokine reactions. Therefore, the study did not encompass an investigation into cytokines.”
Point 6. Did the authors used another inactivation markers other than PD-1?
Response 6: We did not perform other inactivation markers for the phenotypic staining experiments. Our study only focused on the PD-1 expression on different subsets of naïve and memory T cells. PD1 expression was found in early dysfunctional T cells and increased expression in late dysfunctional T cells which was a more severe exhaustion phenotype. Increase expression of PD-1 has been observed in exhaustion phenotype of T cells in viral diseases including COVID-19.
Point 7. Did the authors collected nasal swabs from the individuals and tested for IgA and virus neutralization with swab?
Response 7: During the study period, nasal swabs were collected at the acute respiratory clinic (ARI clinic) and were sent out for treatment purposes, and we could not obtain the fresh samples from the hospital. During the pandemic, hospital’s policies were enforced to prevent the spread of the virus; therefore, we were not allowed to perform a separate research maneuver at the diagnostic counters.
Point 8. Did the authors performed neutralization with new SARS-CoV-2 Omicron variants like XBB.1?
Response 8: While we anticipate conducting such assays, it's important to note that during the study period, the XBB1 variant had not yet become the dominant strain circulating in Thailand. Consequently, we were unable to perform live-virus microneutralization assays due to the unavailability of the virus in our laboratory.